# Applications of Nanozymology in the Detection and Identification of Viral, Bacterial and Fungal Pathogens

**DOI:** 10.3390/ijms23094638

**Published:** 2022-04-22

**Authors:** Sandile Phinda Songca

**Affiliations:** School of Chemistry and Physics, College of Agriculture Engineering and Science, University of KwaZulu-Natal, Durban 4041, South Africa; songcs@ukzn.ac.za; Tel.: +27-(0)31-260-2988 or +27-(0)76-868-7122

**Keywords:** nanozyme, pathogen detection, immunosorbent assay, polymerase chain reaction, enzyme mimic, viral, bacterial, fungal, cancer cell, helminth ova

## Abstract

Nanozymes are synthetic nanoparticulate materials that mimic the biological activities of enzymes by virtue of their surface chemistry. Enzymes catalyze biological reactions with a very high degree of specificity. Examples include the horseradish peroxidase, lactate, glucose, and cholesterol oxidases. For this reason, many industrial uses of enzymes outside their natural environments have been developed. Similar to enzymes, many industrial applications of nanozymes have been developed and used. Unlike the enzymes, however, nanozymes are cost-effectively prepared, purified, stored, and reproducibly and repeatedly used for long periods of time. The detection and identification of pathogens is among some of the reported applications of nanozymes. Three of the methodologic milestones in the evolution of pathogen detection and identification include the incubation and growth, immunoassays and the polymerase chain reaction (PCR) strategies. Although advances in the history of pathogen detection and identification have given rise to novel methods and devices, these are still short of the response speed, accuracy and cost required for point-of-care use. Debuting recently, nanozymology offers significant improvements in the six methodological indicators that are proposed as being key in this review, including simplicity, sensitivity, speed of response, cost, reliability, and durability of the immunoassays and PCR strategies. This review will focus on the applications of nanozymes in the detection and identification of pathogens in samples obtained from foods, natural, and clinical sources. It will highlight the impact of nanozymes in the enzyme-linked immunosorbent and PCR strategies by discussing the mechanistic improvements and the role of the design and architecture of the nanozyme nanoconjugates. Because of their contribution to world health burden, the three most important pathogens that will be considered include viruses, bacteria and fungi. Although not quite seen as pathogens, the review will also consider the detection of cancer cells and helminth parasites. The review leaves very little doubt that nanozymology has introduced remarkable advances in enzyme-linked immunosorbent assays and PCR strategies for detecting these five classes of pathogens. However, a gap still exists in the application of nanozymes to detect and identify fungal pathogens directly, although indirect strategies in which nanozymes are used have been reported. From a mechanistic point of view, the nanozyme technology transfer to laboratory research methods in PCR and enzyme-linked immunosorbent assay studies, and the point-of-care devices such as electronic biosensors and lateral flow detection strips, that is currently taking place, is most likely to give rise to no small revolution in each of the six methodological indicators for pathogen detection and identification. While the evidence of widespread research reports, clinical trials and point-of-care device patents support this view, the gaps that still exist point to a need for more basic research studies to be conducted on the applications of nanozymology in pathogen detection and identification. The multidisciplinary nature of the research on the application of nanozymes in the detection and identification of pathogens requires chemists and physicists for the design, fabrication, and characterization of nanozymes; microbiologists for the design, testing and analysis of the methodologies, and clinicians or clinical researchers for the evaluation of the methodologies and devices in the clinic. Many reports have also implicated required skills in mathematical modelling, and electronic engineering. While the review will conclude with a synopsis of the impact of nanozymology on the detection and identification of viruses, bacteria, fungi, cancer cells, and helminths, it will also point out opportunities that exist in basic research as well as opportunities for innovation aimed at novel laboratory methodologies and devices. In this regard there is no doubt that there are numerous unexplored research areas in the application of nanozymes for the detection of pathogens. For example, most research on the applications of nanozymes for the detection and identification of fungi is so far limited only to the detection of mycotoxins and other chemical compounds associated with fungal infection. Therefore, there is scope for exploration of the application of nanozymes in the direct detection of fungi in foods, especially in the agricultural production thereof. Many fungal species found in seeds severely compromise their use by inactivating the germination thereof. Fungi also produce mycotoxins that can severely compromise the health of humans if consumed.

## 1. Introduction

Biomedical applications of nanoparticles are currently undergoing an exponential increase [1,2,3]. They feature strongly in research aimed at advancing disease diagnosis [4,5], therapy [6,7], and disease prevention through the development of immunity by vaccination [8,9], and by environmental sanitization [10,11]. Because infectious diseases are responsible for many hospitalizations and deaths globally, there has been much attention paid to the applications of nanoparticles in the diagnosis of communicable diseases [12,13]. Of the six classes of pathogenic agents responsible for communicable diseases which include prions, viruses, bacteria, fungi, protozoa, and helminths, much research is establishing new technologies to diagnose infections caused by viruses [14,15], bacteria [16,17], fungi [18,19], and parasites [20,21]. Prions are proteins normally found in the central nervous system, that fold to form an incorrect tertiary structure which cannot perform the normal function of the protein [22,23]. When prions cause correctly folded proteins to undergo tertiary structural reconfiguration and assume the incorrect tertiary structure of the prions, a chain reaction ensues causing an avalanche of proteins to fold incorrectly. As these accumulate within cells, cell death and tissue damage are inevitable. The problem with prions is that they are difficult to detect since they are contained within host tissue cells. Hence, not much is known about them unlike other infectious agents such as viruses, bacteria, fungi, and parasites. The lack of reported research in the literature on prions may be attributed to this as the main factor.

Unlike prions, viral infections are ubiquitous with more than 200 viruses which are known to be able to infect and cause diseases in humans [24,25]. In the majority, viruses comprise a capsid protein or lipid–protein complex encapsulated nucleic material in various shapes and two main classes. Figure 1 shows that the isometric viral classes are more or less spherical within the protein capsid or lipid–protein-encapsulating complex membrane, and that bacteriophages are head-and-tail viruses that infect bacteria by injecting the nucleic material contained in the icosahedral head through the tail. Bacteria are a major cause of morbidity and mortality across the world with the developing world most severely affected [26,27].

Similar to viruses, bacteria are ubiquitous as free-living and mostly unicellular organisms that can aggregate to form colonies. Whereas most of the known bacteria live normally on various parts of the human body without causing illness, some bacterial species are pathogenic. For example, *Escherichia coli* causes food poisoning, while *Helicobacter pylori* and *Neisseria gonorrhoeae* cause gastritis and the sexually transmitted gonorrhea, respectively. The structures of bacteria are diverse, including small spherical, spiral threaded, cylindrical, flagellated rod-like, and filamentous chain-like structures. They are prokaryotes, and therefore lack well-defined nuclei and membrane-bound organelles, but have chromosomes composed of a single closed DNA circle. Virulent bacteria are classified as Gram positive or negative [28,29]. Gram positive bacteria are characterized by retaining the color of the crystal violet stain in the Gram test, because they have a cell wall composed of a thick peptidoglycan layer. Gram negative bacteria on the other hand, do not retain the crystal violet stain in the Gram test, because their cell covering is composed of a thin peptidoglycan wall intervening between an inner cytoplasmic and an outer bacterial cell membrane, both containing permeating membrane proteins.

Fungi often infect the human body through several mechanisms. A virulent fungal species can invade the body and cause disease on the skin, and then spread to other tissues and organs, such as bones, hair, and nails or affect the entire body. Examples of virulent fungal species include *Coccidioides immitis*, *Histoplasma capsulatum*, and *Blastomyces dermatitidis*, which account for many infections each year. Even though some fungal species are normally avirulent in healthy people, they can be transmitted to deep tissues and cause serious infections in patients with compromised immune systems. Examples include *Candida albicans*, *Aspergillus*, *Rhizopus*, and *Fusarium* species [30]. The general structure of the fungal cell wall consists of an outer layer packed with mannoproteins and beta-glucans, with the cytoplasmic membrane forming an inner, interactive layer [31]. The bacterial and fungal walls are shown in Figure 2.

Nowadays the identification and diagnosis of several multicellular organisms which cause disease through infection as parasites can be facilitated through the use of nanotechnology. Parasitic organisms that infect humans which can be identified and diagnosed using nanotechnology include protozoa, helminths, and ectoparasites. Microscopic, single-celled protozoan parasites multiply in humans, and this contributes to their virulence and the serious illnesses that develop from infection even by a single cell. The multicellular helminths on the other hand are generally visible to the naked eye in their adult stages. Examples include worm-like creatures such as *Ascaris lumbricoides*, trematodes, cestodes, and nematodes. The use of nanotechnology in identifying and treating helminth infection has been widely reported [32].

One of the fastest growing applications of nanotechnology today is the identification of pathogens and diagnosis of the diseases they cause, called nanozymology, which is an emerging terminology used to refer to the science of nanozymes [33]. Nanozymes are defined as nanomaterials that act similar to enzymes without most of the limitations of real enzymes; they are cost effectively and exogenously generated to catalyze many biological processes, yet they are stable enough for a long shelf-life storage. For example, iron oxide nanoparticles can act as surrogates to mimic the traditional peroxidase enzymes [34]. Research on nanozymes has increased exponentially over the past decade and with this, the new interdisciplinary area of nanozymology was born [35,36]. While bringing together a wide range of nanozyme applications, properties, mechanisms, and the unique characteristics of metal and metal oxide nanozymes, Liu et al., (2021) provided an elegant review of the catalytic mechanism mimetic activities of a series of metal and metal oxide nanozyme models for several enzyme processes, including the mechanisms of catalase, a unique, stable peroxide decomposition antioxidant enzyme which is also an essential mechanistic partner to the superoxide dismutase enzyme [37]. In addition to the mechanisms of mimetic activities of nanozymes of catalase, peroxidase, oxidase, and superoxide dismutase, Liu et al., (2021) also reviewed in some detail the biosensing applications of nanozymes for the detection of heavy metal ions, antibiotics, antioxidants, and pathogens. Catalytic activities of nanozymes discussed in the current review include the peroxidase reaction which catalyzes the hydrogen peroxide oxidation of substrates such as 3,3′,5,5′-tetramethylbenzidine (TMB), 2,2′-azino-bis(3-ethylbenzthiazoline-6-sulfonic acid) salt (ABTS), and 5-amino-2,3-dihydrophthalazine-1,4-dione, known as luminol.

The current review will focus on the applications of nanozymes for the detection and identification of pathogenic microorganisms and diagnosis of the infectious diseases they cause. Several metal and metal oxide nanozymes have been used for the detection and identification of many microorganisms responsible for infectious diseases. Examples of methods cited by Liu et al., (2021), used for the detection and identification of microorganisms responsible for infectious diseases, include the colorimetric immunoassay, magnetic nanozyme-linked immunosorbent, enzyme-linked immunosorbent, immunochromatographic, colorimetric and electrochemical, nanozyme strip, magnetophoretic chromatography, and the lateral flow essays. Viral pathogens that have been identified and detected using these methods include the murine noroviruses, the viruses that cause the common influenza, the avian influenza, mumps, measles, and listeriosis. Bacterial pathogens include *Bacillus subtilis*, *Streptococcus mutans*, *Pseudomonas aeruginosa*, *Staphylococcus aureus*, *Escherichia coli*, and *Salmonella enterica*. Clearly, Liu et al., (2021) have shown that nanozymology is a fast-growing science in the detection and identification of viral and bacterial pathogens that are known to cause serious illnesses among humans. This review will focus on the details of the methodologies used in this regard, highlight the innovative structure and architecture of the nanoconjugates used as nanozymes, and suggest possible further innovations on the basis of the reported mechanisms and the performance of the nanozymes.

## 2. The Pathogen Challenge

The ubiquitous distribution of virulent pathogenic microorganisms is a major cause of the infections that cause morbidity and mortality across the world. Given that environmental detection has taken the center stage in recent times due to the heightened recognition of the impact of environmental pathogenesis on epidemiology, identification of virulent pathogens in the environment before they cause infection may become more important in the future than the identification for purposes of disease diagnosis. In the developing world, pathogens in the environment are a major driver of epidemics such as cholera [38] and dysentery [39]. Since the onset of the COVID-19 pandemic across the world in December of 2019, environmental transmission has been widely researched through studies which were mainly concerned with survival of the virus in aerosolized droplets and various surfaces [40,41]. With minor differences, most of the studies confirmed aerosol and surface lifetimes that were considered to be long enough to sustain the observed infection rates across the world, supporting the adopted protocols of social distancing [42], wearing of facemasks [43], and environmental sanitization [44]. These studies support the need for more efficient and cost-effective strategies for detecting and identifying virulent pathogens in the environment for the prevention of infections.

The screening of animal vectors such as vermin, pests, and pets and their environments is an essential aspect of the detection and identification of virulent pathogens. For example, much research has supported the zoonotic origins of SARS-CoV-2 [45,46], and many pathogens are now understood to be transmitted through vermin intermediary vectors [47,48]. This reinforces the need for quick and efficient vermin control as well as the accurate and efficient detection and identification of pathogens on vermin and in potential vermin environments [49,50], for the purpose of identifying vermin to be eradicated as well as vermin environments to be sanitized for human health and safety. The dreaded final contact of virulent pathogens with the human body can and often occurs in several ways, such as contact with hands [51,52], and by consumption of infected foods [53,54]. Therefore, there is a case for the detection and identification of virulent pathogens on relevant parts of the human body. Virulent pathogens can multiply and cause disease immediately upon initial contact, or they can be transmitted to parts of the body where their multiplication and disease causing are more likely to be sustained and supported by the external and internal body environment. For example, whereas some fungi will not flourish on dry and exposed skin, they flourish on moist and covered areas such as armpits and foot soles. This supports the case for screening and analysis of various moist and covered areas of the body for specific fungi. While some pathogens are transmitted to internal tissues and organs directly through the skin, others can gain access from food ingestion through the gastrointestinal tract or through compromised skin in the blood stream, making the case for sampling and screening of the blood and gastrointestinal tract for such pathogens.

## 3. Mechanistic Considerations

Several methods are used for the detection of pathogens. Most of these methods are specific for a limited scope of pathogens and are not usable outside their pathogen specificity scope. For example, the enzyme-linked immunosorbent assay is used to detect antibodies of the pathogens that cause specific viral and bacterial diseases [55,56]. It is therefore desirable to determine if the use of nanozymology offers enhanced specificity or any advantages for the detection and identification of virulent pathogens compared to, or when used in conjunction with established methods. The proposed contribution of the current review to science is an analysis of the significance of nanozymology in pathogen detection and identification based on selected literature reports, on the application of nanozymology for the detection and identification of virulent pathogens from various sources. The mechanism of the enzyme-linked immunosorbent assay analysis of pathogen analytes, of evaluating the detection antibody, using the cascade reaction amplification which is typically accomplished using a colorimetric reaction signal is largely emulated among the vast majority of nanozyme methods.

In the same way that the lateral flow immunoassays mimic the enzyme-linked immunosorbent assays on the lateral flow flat pad along which the analyte moves by osmosis, picking up the specific detection antibodies and the signal amplification reagents [57,58], so too do most of the nanozyme lateral flow method types. Nanozymology detection and identification methods therefore use nanozymes generally in place of the enzymes in the traditional methods for the detection and identification of pathogens. In this regard, Sun et al., (2020) cited several noble metal nanoparticles that have been used as nanozymes, including gold, platinum, palladium, and iron, together with a series of the biosensors in which they have been used as enzyme surrogates for the detection and identification of viral and bacterial pathogens [59]. Biosensors play a very important role in point-of-care detection and identification of disease-causing pathogens such as viruses and bacteria for purposes of disease diagnosis and therapy. In most of the point-of-care biosensors currently in use, oxidation by hydrogen peroxide of a variety of substrates using the horseradish peroxidase as the catalyzing enzyme, is replaced with more robust nanozymes such as the noble metal and noble metal chalcogenide nanoparticles [60].

Two of the most widely used enzyme-linked immunosorbent assay tactical approaches include the direct and sandwich strategies. In the direct approach, the antigen and the detection antibody are conjugated directly to form the antigen–antibody conjugate. In the sandwich approach, one antibody is first used to capture the antigen, and this is followed by conjugation with the detection antibody to form an antibody–antigen–antibody sandwich. The detection antibody is always tagged with the detection enzyme. In the case of nanozymology approaches, the detection antibody is always tagged with the detection nanozyme. In some of these studies the nanozymes are augmented with biological material such as enzymes, DNA and RNA strands or fragments [61,62]. However, this tends to diminish the lifetime of the biosensor even though specificity and signal response are enhanced to achieve the desired low limits of detection. The sandwich enzyme-linked mechanism may be illustrated as shown in Figure 3. The primary antibody is used for immobilizing the antibody–antigen conjugate. The secondary nanozyme-linked antibody then conjugates with the primary antibody–antigen conjugate to form the antibody–antigen–antibody sandwich. The nanozyme on the secondary antibody, which is also known as the detection antibody, is responsible for the enzyme mimetic detection reaction which in most cases is colorimetric [63].

Recently, Oh et al., (2020) reported an ultra-sensitive innovation to improve on immunological approaches by using the initial magnetic separation and concentration of the pathogen, followed by nanozyme immunoassay [64]. In their research, the group showed that the enzymelike activity of the gold nanozyme was higher than that of traditional enzymes in the detection and identification of the Influenza A Virus. A clinical trial of this concept of combining magnetic enrichment and nanoenzyme-linked immunosorbent assay signal amplification was recently reported [65] by the same group for the detection and identification of the tuberculosis bacterium, offering an ultra-sensitive, and quick naked eye or plate reader-based point-of-care urine analysis. Due to the potential for low detection limits and design simplicity, this approach has attracted the attention of other researchers for further innovation. For example, Liu et al., (2020) reported magnetic enrichment of *Staphylococcus aureus*, using a bovine serum albumin capped cobalt oxide nanozyme, by attaching the pathogen to the nanozyme using a novel specific fusion phage protein, and isolating the nanozyme–pathogen complex, using magnetophoretic chromatography in an external magnetic field [66]. Ultra-sensitive detection and identification of *Staphylococcus aureus* in milk was achieved.

## 4. Recent Approaches of Pathogen Detection

Due to the multi-stage value chain, from the farm to retail sales, along which the vast majority of food contamination with virulent pathogens often occurs, foods have great potential to become sources of virulent pathogenic infections to humans [67]. Traditional approaches of detection of virulent pathogens are expensive, time-consuming, and suffer from high detection limits and low pathogen specificity. Some of the most widely used traditional methods include the nucleic acid-based processes such as PCR, the nucleic acid sequence-based amplification, the loop-mediated isothermal amplification, and the oligonucleotide DNA microarray, all of which are multi-step, time consuming, and costly procedures [68]. Key steps of the PCR-based procedures of pathogen detection and identification include the initial isolation of the pathogen, followed by the extraction of its DNA, denaturing, priming, DNA polymerase enzyme catalyzed amplification, and amplicon analysis, usually by means of gel electrophoresis based procedures [69]. For example, a novel nanofluidic real-time PCR system developed for qualitative and quantitative molecular detection and identification of 17 human and 2 food borne viruses performed marginally better than the conventional real-time manual PCR system. The procedure involved a 55 °C reverse transcription (60 min), a 95 °C denaturation (15 min), 65 °C amplification (15 min × 45 cycles), 60 °C (1 min), and 65 °C (1 min) [70].

A traditional approach for the detection and identification of bacteria in foods may be illustrated with the real-time PCR detection of 12 common food-borne bacterial pathogens reported by Liu et al., (2019). Their procedure consisted of a 95 °C denaturation (30 min), 95 °C amplification (5 s × 40 cycles), 55 °C (10 s), and 72 °C (30 s) [71]. Whereas bacterial specimens obtained from human external environments such as skin, clothes, and personal effects may be sampled and treated the same way as food borne samples, clinical specimens require an entirely different set of sampling procedures such as those described by Yamamoto (2002) [72], followed by PCR procedures of the primed nuclear material fragments. For example, Järvinen et al., (2009) developed a PCR method for the detection and identification of bacteria and tested it on 186 blood culture samples. Their optimized procedure consisted of a 95 °C denaturation (15 min), 96 °C amplification (10 s × 36 cycles), 52 °C (35 s), 72 °C (10 s), 96 °C (5 s × 5 cycles), and 68 °C (30 s) [73]. Before the PCR methods became the standard, traditional techniques for detection and identification of foodborne fungi were based on incubating and growing the sampled fungi on growth media in the Petri dish [74]. Nowadays, however, the PCR methods are highly preferred [75]. For example, fungi on carrot seeds were identified by a PCR method after DNA extraction, using 94 °C denaturation (10 min), 94 °C amplification (1 min × 36 cycles), 58 °C (1 min), 72 °C (1.5 min), and 72 °C (10 min) [76].

Responding to rising fungal infections and the need for fast and reliable detection and identification of the responsible pathogens, Wagner et al., (2018) reported a comparative evaluation of the PCR analysis of the small ribosomal unit DNA, the internal subscriber region of the ribosomal DNA, and the culture method. Using the same PCR parameters, they found a high degree of congruence between the three procedures although the PCR methods were significantly faster and more reliable [77]. The response time of all three methods, however, was a matter of days, with the culture method requiring up to 3–7 days compared to the small ribosomal unit DNA method which was done in one day. While supporting the significant improvement of response speed due to the move to the PCR methods compared to the incubation and growth methods which predominated before the PCR methods, these results still show the need for improvement of the PCR and the immunoassay response speed.

## 5. Nanozymology

Recently developed nanozyme strategies for detecting and identifying foodborne viral pathogens have presented an opportunity for developing ultrafast and cost-effective assays, taking the limits of detection much lower than those of the culture-and-grow, the traditional immunoassays, and the traditional PCR strategies. Although sensitively and reliably detected using the reverse transcription quantitative PCR methods, Weerathunge et al., (2019) proposed a novel colorimetric nanozyme strategy for fast and ultrasensitive detection and identification of the foodborne human norovirus responsible for gastroenteritis outbreaks worldwide. The approach combines the nanozyme activity of AG3-aptamer conjugated gold nanoparticles with the high binding specificity of the aptamer for the surrogate human norovirus, to develop a blue color producing sensor in response to this human norovirus surrogate, in which the blue color intensity is directly proportional to the virus concentration [78]. The sequence of the AG3 aptamer (G-C-T-A-G-C-G-A-A-T-T-C-C-G-T-A-C-G-A-A-G-G-G-C-G-A-A-T-T-C-C-A-C-A-T-T-G-G-G-C-T-G-C-A-G-C-C-C-G-G-G-G-G-A-T-C-C) used by Weerathunge et al., (2019) was discovered by Giamberardino et al., (2013) using the SELEX procedure [79] (Figure 4). Aptamers are short single strands of nucleic or xeno nucleic acid sequences that are prepared and used for their high binding or interaction affinity, selectivity and specificity for various molecular and microorganism targets [80]. The target specificity of aptamers derives primarily from their tertiary structures which consist mainly of their helical single stranded loops and nucleotide base-paired portions [81,82]. As a result, aptamers are increasingly used in biosensing and therapeutic platforms [83]. The value of the aptamer coated gold nanozyme that was recognized by researchers is that several target-specific aptamers that form an aptamer layer around the gold nanozyme via non-specific binding are removed from the gold nanozyme outer aptamer layer by specific binding to the analyte target, exposing the nanozyme surface for its colorimetric nanozyme activity [84,85].

The gold nanoparticle nanozyme activity produces an intense blue color upon interaction with the human norovirus. The AG3-aptamer conjugated nanozyme loses this property because the gold nanoparticles are concealed under the shell of the AG3-aptamer. The surrogate human norovirus, however, replaces the aptamer shell, thus restoring the blue color production activity of the nanozyme in a norovirus concentration linearly dependent manner, thus creating an ultrasensitive sensor for the norovirus. Other pathogens do not bind to the human norovirus specific aptamer. Therefore, the sensor is highly specific for the human norovirus.

A novel graphene quantum dot (GQD) nanozyme immunosensor strategy was reported as part of an electrochemical sensor that was developed as an efficient method for the detection and identification of the virulent bacterial pathogen *Yersinia enterecolitica*, which is responsible for the symptoms of yersiniosis. The immunosensor detected *Yersinia enterecolitica* in milk to the ultralow detection limit of 5 CFU/mL [86]. The gold working electrode was treated with GQDs, followed by antibody immobilization using 1-ethyl-3-(3-dimethylaminopropyl)-carbodiimide, which was then treated with bovine serum albumin and ethanolamine to inactivate carbonyl groups and eliminate any possible non-specific analyte detection. *Yersinia enterecolitica* immobilized on the electrode inhibits the nanozyme redox current, and this means that the detection signal is inversely proportional to the bacterial analyte concentration in the sample. The preparation of the working electrode is shown in Figure 5.

## 6. Indirect Nanozyme-Based Fungal Detection

While literature is lacking on nanozyme strategies for direct fungal detection and identification, similar to natural enzymes, nanozymes have been implicated in the strategies for the destruction of fungal pathogens [87]. Given the recently reported literature suggesting significant enhancement of the detection limits, speed of response, reliability, accuracy, specificity, reproducibility, and shelf-life, of innovations involving nanozymes for the detection of viral and bacterial pathogens, it is well worth exploring how some of these reported methods can be emulated in the detection and identification of fungal pathogens. However, a nanozyme detection of the fungal extract zearalenone (Figure 6), an estrogenic metabolite produced by the fungal species *Gibberella zeae*, has been reported [88]. The gold nanoparticle nanozyme activity produces an intense blue color in the presence of TMB. This approach combines the nanozyme activity of gold nanoparticles with the high zearalenone specificity of the zearalenone specific aptamer in that the aptamer capped gold nanoparticles which initially lose their nanozyme activity, regain it as the zearalenone replaces the aptamer layer over the gold nanoparticles, and the zearalenone proportionate blue color intensity development can be observed with the naked eye. A layer of the zearalenone specific aptamer forms around the gold nanoparticles by weak non-specific binding, thus, blocking its peroxidase activity and conversion of the blue color production in the presence of TMB. Upon introduction of zearalenone, however, stronger specific binding with the aptamer leads to stripping of the capping aptamer layer, exposure of the nanozyme, and unblocking of its peroxidase activity based blue color production. Thus, the visible blue color intensity is linearly proportional to the zearalenone concentration.

Although they cannot really be described as nanozymes, metal-organic frameworks, which are going through phenomenal growth in research studies and applications, also mimic a number of the same enzymes that nanozymes are known for mimicking. For this reason, they are also used in the same nanozyme-based bio-analytical procedures as nanozymes, notably for the detection and identification of pathogens. A variant of the enzyme-linked immunosorbent assay that uses Fe-MIL-88B–NH_2_ in the place of the traditionally used enzymes, was developed for the detection and identification of aflatoxin B1 [89]. Fe-MIL-88B–NH_2_ which was used to mimic the peroxidase enzyme activity in this study, is a metal-organic framework that is made up of ferrous ions linked with 2-aminobenzene dicarboxylic acid [90]. The aflatoxin B1 analyte which was used as the target antigen in this study, is a carcinogen that is produced by some Aspergillus fungal species. Therefore, the method could be used as an indirect detection and identification of *Aspergillus* species in foods by a metal-organic framework-linked immunosorbent assay. The results of the study showed a 90% sensitivity and a 96% specificity which was an improvement on the measured 63% sensitivity and 90% specificity of the enzyme-linked immunosorbent assay used for comparison in the study.

## 7. Direct Nanozyme-Based Cancer Cell Detection

Although cancer cells are not explicitly included in the strict definition of pathogens, once they are formed in the body, they cause cancerous tumors, and they spread across the body as part of cancerous tumor metastasis. The formation of cancer cells including the metastasis processes are included in what is commonly referred to as cancer pathogenesis [91,92]. For this reason, this review includes a discussion of nanozyme mediated detection and identification of cancer cells. A recent review by Ma et al., (2021) provides an impressive list of reported methods developed for the direct detection and identification of cancer cells [93].

One of these methods is a direct detection and identification of cancer cells using graphene oxide decorated with nanoparticles of platinum which are functionalized with folic acid [94]. The method takes advantage of the over-expression of folate receptors on cancer cells [95], the hypoxia mediated over-production of hydrogen peroxide in the tumor and cancer cell microenvironment, and the peroxidase enzyme mimicry of the folic acid functionalized graphene oxide which is decorated with platinum nanoparticles. Treating cancer cells with folic acid functionalized platinum nanoparticle decorated graphene oxide leads to extensive binding of the conjugate on the cells. In the presence of the TMB and hydrogen peroxide, the nanoconjugate catalyzes production of the blue colored single electron oxidation product from TMB, which is visible to the naked eye in a cancer cell specific, and sensitive manner. The method afforded visual colorimetric detection down to 125 MCF-7 cells without any false positives or negatives, indicating a high degree of specificity.

Additionally, a copper oxide nanozyme supported on a nanocomposite consisting of graphene oxide decorated with gold nanoparticles was used to amplify the electrochemical signal for the detection and identification down to 27 circulating MCF-7 cells using cyclic voltammetry [96]. The cyclic voltammetry working electrode was assembled by overlaying a highly polished and clean glassy carbon electrode already treated with the nanogold decorated graphene oxide with the mucin-1 aptamer, to take advantage of the over-expression of the MCF-7 specific mucin 1 protein on the MCF-7 cells. The electrode is then treated with the nanozyme and incubated before cyclic voltammetry measurements as shown in Figure 7.

## 8. Direct Nanozyme-Based Parasite Ova Detection

The detection of parasites appears to be mainly dependent on the indirect methodologies such as the detection of parasite ova in the water using techniques such as (i) graphic enumeration using a microscope, (ii) PCR-based techniques, and (iii) flow cytometry [97]. These techniques are expensive and take quite some time to conclude. As an effective and accurate way of detecting and identifying helminths in waste waters, using PCR, the laboratory manual procedure was recently improved by the development of a recombinase polymerase amplification mechanism in a lateral flow strip [98]. However, a few research reports have emerged recently in which nanozymology is used to detect parasites directly. For example, the detection and identification of *Leishmania* parasites using a chronoamperometric procedure in which nanogold is deposited on the working electrode to act as the catalyst for the conversion of hydrogen ions to hydrogen, while monitoring the faradaic current as the detection signal, has been reported [99]. This approach exploits the nanozyme capability of gold nanoparticles for the hydrogen formation reaction and the specific binding of casein with the zinc metalloprotease leishmanolysin found on the surface of the parasite. Therefore, incubation of casein conjugated gold nanoparticles results in specific binding of the *Leishmania* parasites.

The pellet obtained from the incubation of casein capped nanogold and centrifuge isolation is analyzed using a chronoamperometric procedure on a screen-printed carbon electrode. This procedure proved to be sensitive down to a single parasite per milliliter, with a linear dependence on *Leishmania* parasite concentration over a 7-log range. In addition, a nanogold-based enzyme-linked immunosorbent assay was used for the detection and identification of *Trichinella spiralis*, in which the horse radish peroxidase enzyme-linked immunosorbent assay was used alongside a nanozyme-linked immunosorbent assay, wherein the horse radish peroxidase enzyme was replaced with gold nanoparticles as the nanozyme, and polyclonal antibody immunoglobulins were used to capture and immobilize the *Trichinella spiralis* [100].

## 9. Bacterial Detection and Identification

The nanozyme platform shown in Figure 8 using hybrid nanoflowers of hemin and concanavalin A was reported to detect *Eschericia coli* very sensitively by efficiently mimicking the peroxidase enzyme activity, employing the sandwich type immunoassay strategy [101]. In this research the magnetic beads tagged with polyclonal antibodies were used as the primary antibodies whereupon the bacteria are entrapped, while the hybrid nanoflowers of hemin and concanavalin A were used as the detection or secondary antibody surrogates. Thus, an antibody–bacterium–nanoflower sandwich conjugate was formed and separated magnetically, enriching the analyte before analysis using the nanozyme peroxidase conversion of the colorless divalent ABTS diammonium salt to its green monovalent product. The method showed significant selectivity for *Eschericia coli* compared to other bacteria, with remarkable sensitivity down to 10 bacteria per milliliter.

## 10. Pathogen Biosensors

Biosensors for pathogen detection and identification vary in complexity, sensitivity, speed of response, cost, reliability, and durability, which are considered to be among the most desirable indicators. Colorimetric paper-based biosensors rival most in this regard especially when coupled with color amplification strategies [102]. The three main variants of paper-based biosensors include the dipstick or spot-test assay, lateral flow assay, and paper-based microfluidic analytical devices [103,104]. However, these too vary widely in the six key indicators. While spot or dip test paper strips can be the cheapest, simplest, and quickest, they may lack quantitative sensitivity. Lateral flow assay devices on the other hand have been taken to extreme levels of sophistication by the introduction of digital signal display [105,106], while microfluidic devices offer rapid diagnosis of small samples and reagents, reproducibility, and are very easy to use [107,108].

The performance of many pathogen biosensor devices can be substantially improved by using nanozymes in their biosensor mechanisms. Nanozymes can be used to detect pathogens directly by mechanisms that involve the direct interaction of nanozymes with the pathogen, or indirectly by mechanisms that involve interaction of nanozymes with chemical compounds produced by the pathogen. For example, the indirect detection of bacteria in urine will surely be improved by incorporation of the results of the many reports on rapid and selective detection of nitrite, based on nanozymes because the nitrite is one of the products of bacterial infection [109,110]. The direct detection and identification of bacteria will also be positively impacted by the increasing number of recent reports on nanozyme-based detection and identification of bacterial pathogens. An example of a nanozyme-based technology that holds great promise to improve the direct detection and identification of *Salmonella typhimurium* with high sensitivity is found in a recent report on a sensor based on polymer nanospheres decorated with platinum quantum dots [111], which uses the sandwich immunoassay strategy. *Salmonella typhimurium* specific capture antibodies immobilized on magnetic beads are initially used to enrich the analyte magnetically before conjugation with the detection antibodies that are tagged with platinum quantum dot decorated polymer nanospheres. The platinum quantum dot decorated polymer nanosphere nanozyme mimics the peroxidase conversion of the TMB to form its blue colored single electron oxidation product visible to the naked eye in a highly *Salmonella typhimurium* specific, sensitive, and pathogen analyte linearly dependent manner. This mechanism is shown in Figure 9.

Another example appearing in a recent review is the use of photothermal immunoassay signal transduction [112]. In this colorimetric photothermal immunoassay, the sandwich strategy is used to incorporate the peroxidase mimicking iron oxide nanoparticles attached to the detection antibody in the capture-antibody–pathogen-detection–antibody sandwich conjugate, to convert the TMB to the blue colored one-electron oxidation product, which has a high photothermal conversion when irradiated with an 808 nm laser. Therefore, the detection of the pathogen antigen emanates from two immunoassay signals; the color development and the temperature elevation upon laser light irradiation at 808 nm, both of which are directly proportional to the pathogen concentration [113]. The photothermal immunoassay signal transduction is shown in Figure 10.

## 11. Microfluidic Biosensors

Microfluidic biosensors hold great promise for the detection and identification of pathogens at the point-of-care, for general and bio-analytical research, and biomedical engineering research laboratories because of the possibilities offered by the multiple channel miniature fluidic flows. A microfluidic system consists of multiple microfluid flow channels to enable virtually any of the operational steps that can be rationally imagined and designed into chemical and biological analytical procedures to take place in a miniature chip that can be as small as the tip of a finger. Microfluidic chips are constructed by elaborate carving of the microfluid flow channels, microvalves, and micro-fluid flow pumps [114]. The microfluid biosensor can incorporate electrical [115], thermal [116], magnetic [117], photonic [118], and other physical micro-devices in line with the design requirements analysis. Microfluidic biosensors have attracted much attention over recent decades because of their potential for time and reagent savings [119] and also because they present an innovative new analytical ideology which can potentially be extensively computerized [120,121] to very high precision analytical capabilities.

In recent times, microfluidic systems have been advanced by incorporation of nanozymology into their design and analytical strategies for the detection and identification of pathogens in food, environmental and clinical samples, affording timely analytical results. Firstly, a colorimetric microfluidic biosensor methodology was reported recently by Xue et al., (2021) which utilizes the magnetic analyte enrichment sandwich immunoassay strategy and measurement of *Salmonella* analyte concentration, based on the color development of the nanozyme-based peroxide oxidation of TMB. The immuno-magnetic nanoparticles formed by mixing of capture antibodies with the magnetic nanoparticles followed by addition of the *Salmonella* sample are treated with immuno-nanozyme conjugate obtained by the detection antibodies and the manganese dioxide nanoflowers. The foregoing magnetic immunoassay conjugate reagents are thoroughly mixed in the spiral micromixer microfluid incubation chip and then magnetically captured in the separation chamber of the microfluid chip as shown in Figure 11.

Addition of TMB causes the colorimetric reaction catalyzed by the terminal manganese nanoflowers on the magnetically captured sandwich conjugate to give the colored single electron oxidation product. In this microfluidic immunoassay strategy, the *Salmonella* analyte concentration data collected from the detection chamber of the chip are captured, processed, and displayed on a smartphone [122].

In the second example, a fluorescence microfluidic biosensor methodology was reported by Hao et al., (2020) which also utilizes the magnetic analyte immobilization sandwich immunoassay strategy, but the measurement of the analyte concentration is based on the fluorescence intensity of the released quantum dots [123]. The immuno-nanozyme conjugate formed by conjugation of the polyclonal antibodies with the manganese nanoflowers is conjugated with the *Salmonella* specific monoclonal antibody conjugated immuno-magnetic beads, which capture the bacteria to form the sandwich conjugate with the Salmonella analyte. In this study, the amino-modified manganese dioxide nanozyme is conjugated with carboxyl-modified fluorescent quantum dots as the fluorescence signal generators via the 1-ethyl-3-(3-dimethylaminopropyl)-carbodiimide. The analyte, immuno-magnetic beads, and the immuno-nanozyme conjugate are injected together into the microfluidic chip for mixing and formation of the nanoconjugate consisting of the *Salmonella* specific monoclonal antibody immunomagnetic beads, the *Salmonella*, and the fluorescent quantum dot decorated polyclonal antibody immuno-nanozyme. This is achieved by thorough mixing and incubation in the microfluidic chip. After release of the quantum dots by glutathione dissolution of the manganese dioxide nanoflowers, the fluorescence of the quantum dots is measured to quantify the *Salmonella*. This fluorescence microfluidic biosensor methodology was reported to achieve a detection limit of 43 colony forming *Salmonella* per milliliter, from chicken meat derived samples. The microfluidic chips of these colorimetric and fluorescence microfluidic biosensors are illustrated in Figure 12.

## 12. The Paper Dipstick or Spot Test Methods

The attractiveness of paper strip tests is their simplicity, response speed, low cost, and ease of use, characteristics that make them the preferred point-of-care test methods in such resource constrained environments such as the developing world, especially in their rural areas. A nanozyme strip was developed for rapid local diagnosis of *Ebola*, a deadly pathogen that is raging in West Africa without cure [124]. The indirect detection nanozyme test-strip is 100 times more sensitive than the standard colloidal gold nanoparticle strip such as the one reported by Sheng et al., (2020) [125]. It is capable of detecting down to 240 plaque-forming units and is therefore comparable to the standard enzyme-linked immunosorbent assays in response time, giving a reliable result in less than half an hour.

A test line consisting of the *Ebola* capture antibody 1H3 in borate buffer and a control line consisting of goat anti-mouse IgG antibody in borate buffer were dispensed on an absorbent pad terminated nitrocellulose membrane strip as shown in Figure 13. The test line captures the glycoprotein of the Ebola virus sandwiched between the capture antibody 1H3 and the iron oxide nanoparticle nanozyme tagged detection antibody. The iron oxide nanozyme attached to the detection antibody produces the colorimetric peroxidase mimic conversion of TMB to form its blue colored single electron oxidation product visible to the naked eye in an Ebola specific, and sensitive manner. In this research, the nanozyme strip developed for detection and identification of Ebola was compared to the standard strip made of colloidal gold nanoparticles. The gold and iron oxide test strips are illustrated in Figure 13.

## 13. Multiple Flow Rate Multiple Channel Colorimetric Paper Strip

The rate of movement of water by diffusion across the nitrocellulose membrane used in paper-based dipstick methods is generally inversely proportional to the density of the membrane. With this in mind, the variable pressure pressing of the nitrocellulose membrane introduced for the first time by Park et al., (2016) in paper-based dipstick research methodology, achieved a variable fluid flow rate due to the different density regions created on the nitrocellulose paper membrane, and demonstrated the intended impact on the sequential multi-step immunoassay reactions that take place at predetermined positions, where the various immunoassay reagents are immobilized on the nitrocellulose membrane paper strip [126]. For this strategy to work, all the required immunoassay reagents are either immobilized or localized in different parts of the nitrocellulose membrane paper strip, according to the desired sequence of reactions and their colorimetric response. This strategy also enabled the creation of multiple fluid flow channels for the detection and identification of different analyte pathogen constituents of the sample. This was achieved by dividing the nitrocellulose membrane paper strip into parallel flow regions as shown in Figure 14, using high-density region lines along the paper strip created by applying high pressure, were used as the channel dividers, and different pathogen specific immunoassay reagents were immobilized in each test line.

In this research, the sandwich type immunoassay methodology was used by locating the capture antibodies at the test line after the nanozyme tagged detection antibodies in the water diffusion direction, so that the water hydrates and mobilizes the nanozyme tagged antibodies with the pathogen, before binding the capture antibody at the test line, at which stage the desired pathogen specific nanogold tagged sandwich antibody–pathogen–antibody conjugate is formed. When the pathogen specific nanogold tagged antibody conjugate reaches the test line, the nanogold generates the amplified color signal, which is visible to the naked eye, thus enabling detection and identification of each pathogen in its specific channel test line. The slow flow channels 1 and 3 of the paper strip enable amplification of the test line colorimetric signal by generating gold ions which attach on the gold nanoparticles attached to the detection antibodies of the nanogold tagged sandwich antibody–pathogen–antibody conjugate. The paper strip working principle is illustrated in Figure 14 showing the different flow rate multiple channels.

Although the detection limits of the paper strip were still very high and the plan was to further reduce them by optimization of the methodology, the current value proposition of the multiple flow rate and multiple channel colorimetric paper strip reported for the first time was the simultaneous analysis of two different microorganisms. This was demonstrated by the simultaneous analysis of *Escherichia coli* and *Salmonella typhimurium* to the detection limits of 10^5^ and 10^6^ CFU/mL, respectively.

## 14. Chemiluminescence Nanozyme-Linked Immunoassay Paper Strip

Whereas the different flow rate multiple channels amplified colorimetric nanogold paper strip reported by Park et al., (2016) enhanced the nanogold signal to achieve superior detection and identification of *Salmonella typhimurium* and *Escherichia coli* by the naked eye, much like the nanogold based nanozyme-linked immunoassay paper strips, colorimetric chemiluminescence paper strips present an elegant way to amplify the naked eye visualization of the colorimetric immunoassay detection signal. A paper strip based on an iron oxide magnetic nanoparticle nanozyme-linked immunoassay significantly enhanced the detection and identification of the SARS-CoV-2 spike protein receptor-binding domain derived antigen by visualizing the chemiluminescence signal obtained from nanozyme catalytic conversion of luminol [127]. In this immunoassay strategy, the iron oxide hemin core-shell nanoparticle based nanozyme mimics the horse radish peroxidase enzyme.

A sandwich type immunoassay method is used. The sample containing the SARS-CoV-2 spike protein derived receptor-binding domain antigen binds with the nanozyme tagged capture antibody, to form a sandwich antibody–antigen–antibody conjugate upon binding with the capture antibody immobilized at the test line in the presence of luminol, whereupon the nanozyme tagged to the conjugate catalyzes the chemiluminescence conversion of 5-Amino-2,3-dihydrophthalazine-1,4-dione or luminol. In this strategy, the chemiluminescence is captured by camera or cell phone camera which can process the signal intensity using an appropriate mobile application for elegant visual display and graphic analysis. The arrangement and working principle of the strategy is illustrated in Figure 15, showing the sampling pad and the conjugation pad, the test line, and the control line, along the fluid flow direction. Due to the nanozyme linked immunoassay chemiluminescence signal enhancement, the paper strip detects down to a median tissue culture infectious dose of 360 pathogens per milliliter, in 16 min or less.

## 15. Future Implications with Therapeutics and Applications

Because the impact of the novel nanozyme based innovations on the detection and identification of disease-causing pathogens is shorter detection time, lower detection limits, cost-effective detection, concentration dependent results, and quality devices, point-of-care usage will be enhanced. These improvements in detection and identification of disease-causing pathogens at the point-of-care will benefit accurate diagnosis and responsive therapeutic strategies. Kumawat et al., (2021) predict that nanozymes will play a leading role in the fight against the current and future pandemics [128]. Along with the rise in nanozyme-based diagnostics, several novel nanozyme-based therapeutic technologies are already enhancing the therapeutic strategic possibilities in the clinic [129,130]. The combined impact of new technologies in nanozyme-based diagnostics and therapeutics is improvement of healthcare and disease burden [131]. This is already evident in the impact of nanozymes in the diagnosis and treatment of COVID-19 [63]. Additionally emerging recently is the nanozyme technology of genetic modification by splicing, editing, knockdown, and knockout, which has enabled transient protein variety and genetic disease therapy [132]. For example, a gold nanozyme functionalized with two catalytic DNA strands and an RNA ligase was reported recently as capable of splicing virtually any RNA stem-loop [133]. Through their applications in detecting and identifying pathogens in water and foods, nanozyme-based detection of pathogens in the environment is starting to be among the leading strategies for management of epidemics [78,134]. It is also playing a leading role in the detection of pesticides and other pollutants in the environment [135].

## 16. Conclusions

While the detection and identification of pathogens has evolved from the historic culture-and-grow methods, through the laboratory-based manual analytical procedures, the rise in pathogen sponsored disease burden, suffering, and deaths over the evolution period, has escalated the need for fast and cost-effective point-of-care methods, giving rise to many paper-based and microfluidic methods of detection and identification of pathogens. Significantly during this period, the PCR and immunoassay methods have contributed many significant gains in some of the most important bio-analytical methodology and strategy characteristics, including simplicity, sensitivity, speed of response, cost, reliability, specificity, and durability. Because many of these methods involved elaborate use of natural enzymes, which are expensive and may suffer from gradual degradation, significantly compromising their shelf life and reliability, the advent of nanozymes, which mimic the catalytic activities of natural enzymes, has brought many improvements in these methodological characteristics. However, nanozymes still lack specificity for pathogens and for these reasons they have been used in combination with methods such as PCR, enzyme-linked immunosorbent assays, paper strips, and microfluidic systems which are pathogen specific.

This review presents a synopsis of the details of some of the key methods that have been developed and tested for pathogen detection and identification. To put these developments in perspective, a discussion of the types, classes, structures of pathogens, and the implicated disease burden led to an analysis of the historic and current methods used for pathogen detection and identification. In these discussions, it was necessary to provide some detail to rationalize the developmental trajectories based on the difficulties and challenges of traditional approaches along the six most desirable methodological indicators that are proposed as key in this review. While the culture-and-grow method still predominates in resource constrained environments due to lack of infrastructural and knowledge resources, an increasing number of paper-based methods are coming in to ease the burden. For example, it is still considered to be quite normal and robust that sampling in resource constrained environments is immediately followed by sending samples away for analysis and the waiting periods are major drivers of the disease burden and mortality. This is the area in which paper strips are providing the most needed relief [136,137].

Because the application of nanozymology is the central theme of focus, the review went into detailed analysis of applications of nanozymes in viral, bacterial, and fungal pathogens, while indicating some of the applications in cancer cell and parasite detection. In this regard, examples of how nanozymes have been used in colorimetric and photothermal colorimetric nanozyme-linked immunoassay detection and identification of bacteria were discussed. Other examples discussed in some detail include a colorimetric microfluidic and a fluorescence microfluidic biosensor, an iron oxide nanozyme colorimetric immunoassay paper strip, and a nanozyme-linked immunoassay chemiluminescence paper strip. The two paper strips were discussed with the background of an innovative different flow rate multiple channel amplified colorimetric nanogold paper strip. The immunoassay paper strip and microfluidic biosensor methodologies are increasingly utilizing the capabilities of real time smart phone data capture and display.

Due to their importance emerging from the discussions, a comparative analysis of the various methodologies for detection and identification of pathogens considered in the evolutionary profile presented in this review was conducted along the set of six desirable characteristics—complexity, sensitivity, speed of response, cost, reliability, specificity, and durability. The methodological classes considered in this review include culture-and-grow, manual laboratory, and biosensors. Manual laboratory methods include the enzyme-linked immunosorbent assays and PCR. Biosensors include lateral flow paper strips and microfluidic devices. Whereas Li et al., (2000) recognized target recognition molecules, signal transduction, and the substrate material as the three key functional components of biosensors, the nanozymology biosensors considered in this review focused on pathogens directly and molecular antigens indirectly, the colorimetric, photothermal, chemiluminescence, photoluminescence, surface plasmonic, and fluorescence signal transduction, based on nanozymes mimicking enzymes [138].

It is clear that nanozymology has taken the detection and identification of pathogens to greater heights in terms of the six characteristics identified in this review, including specificity.

## Figures and Tables

**Figure 1 ijms-23-04638-f001:**
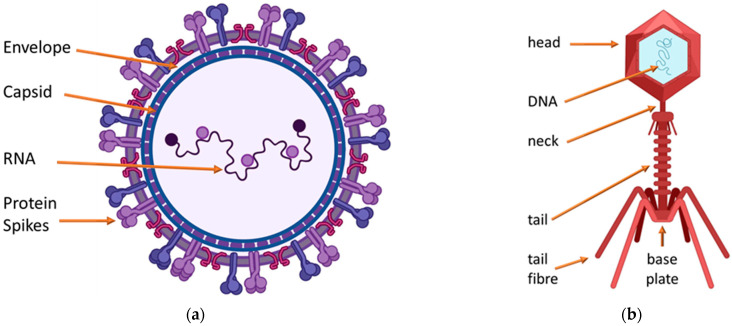
There are two major morphological classes of viruses, the isometric class and the bacteriophages. (**a**) Isometric viruses are more or less spherical, (**b**) macrophages are head-and-tail viruses.

**Figure 2 ijms-23-04638-f002:**
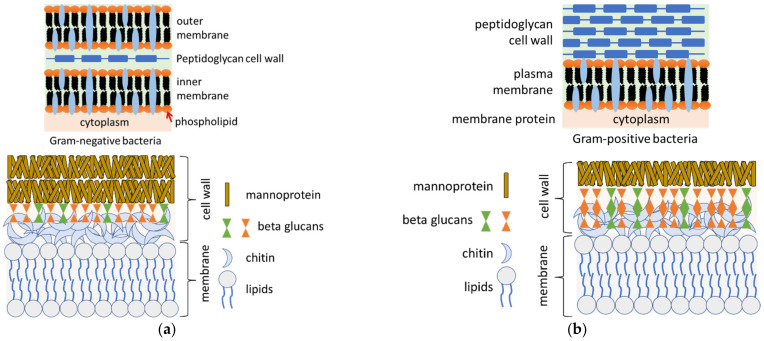
Illustration of the structure of the cell walls of Gram negative bacteria, Gram positive bacteria, and fungal cell walls showing (**a**) hyphae cell wall and (**b**) yeast cell wall.

**Figure 3 ijms-23-04638-f003:**
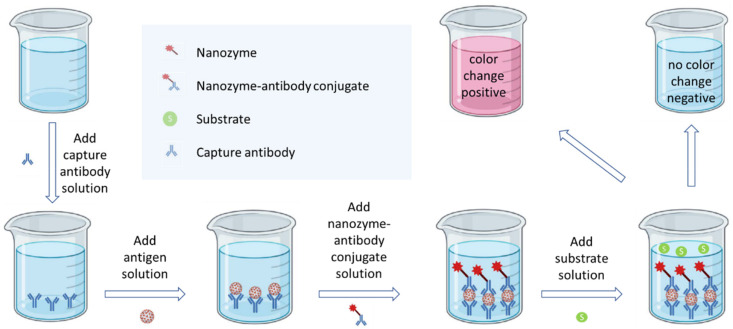
Graphical illustration of generic microbial nanozyme-linked immunoassay methodology.

**Figure 4 ijms-23-04638-f004:**
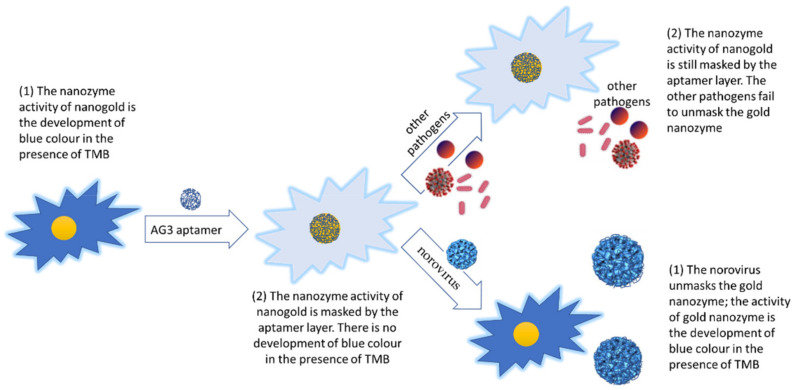
Graphical illustration of the novel colorimetric nanozyme strategy for detection and identification of the foodborne human norovirus.

**Figure 5 ijms-23-04638-f005:**
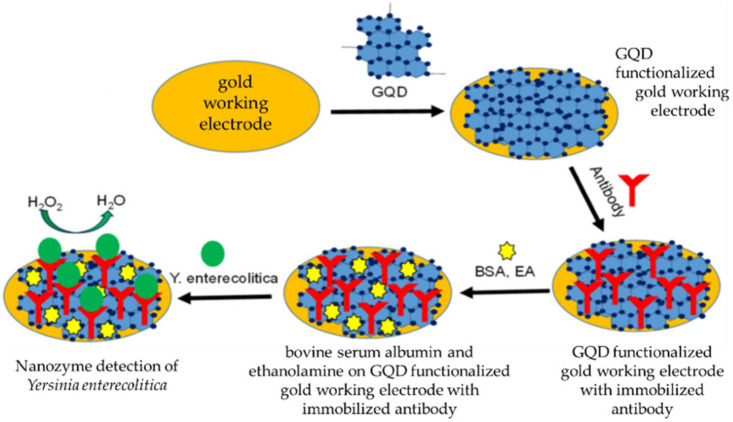
Preparation of the GQD working electrode of the immunosensor (reproduced with minor modification from Savas et al., (2019) [77] under the creative commons attribution license 4.0).

**Figure 6 ijms-23-04638-f006:**
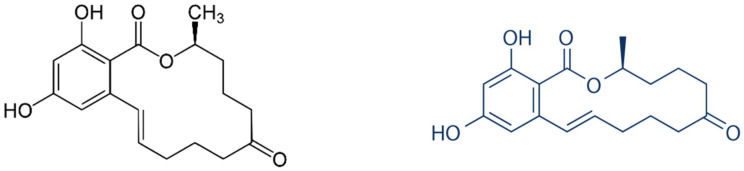
Chemical structures of the fungal extract zearalenone, an estrogenic metabolite produced by the fungal species *Gibberella zeae* (Sun et al., 2018) [88].

**Figure 7 ijms-23-04638-f007:**
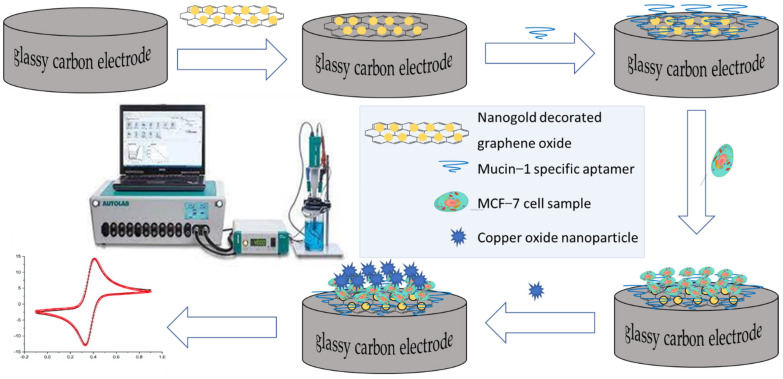
Illustration of the cyclic voltametric detection of circulating MCF-7 cells by means of a copper oxide nanozyme supported on a nanocomposite consisting of graphene oxide decorated with gold nanoparticles.

**Figure 8 ijms-23-04638-f008:**
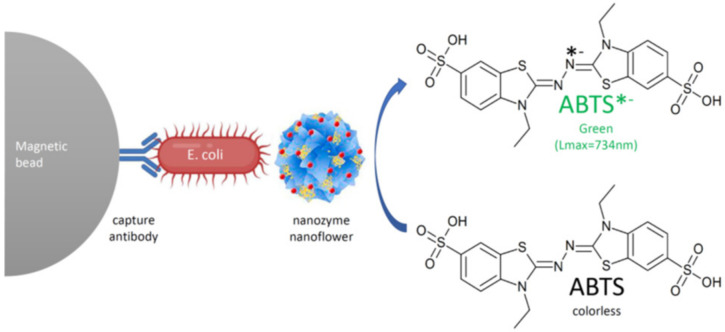
Colorimetric reaction of the peroxidase mimicked by the nanozyme nanoflowers of hemin and concanavalin A showing the colorimetric conversion of 2,2′-azino-bis (3-ethylbenzthiazoline-6-sulfonic acid) from colorless to green as the detection signal.

**Figure 9 ijms-23-04638-f009:**
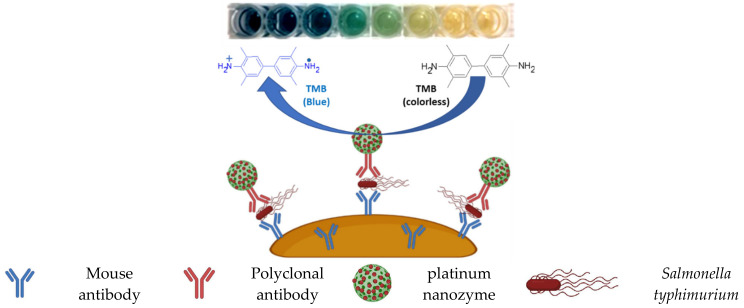
Colorimetric detection and quantification of *Salmonella typhimurium* (redrawn from Hu et al., 2021).

**Figure 10 ijms-23-04638-f010:**
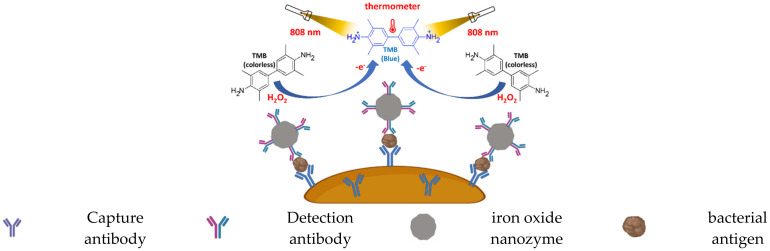
Illustration of the colorimetric photothermal nanozyme-linked immunoassay (redrawn from ref Alizadeh et al., 2021).

**Figure 11 ijms-23-04638-f011:**
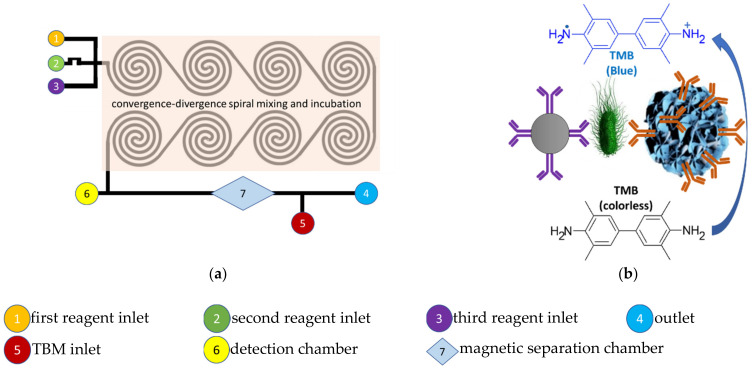
Illustrations of the nanozyme enhanced microfluidic system of Xue et al., (2021) [122]. (**a**) The nanozyme enhanced microfluidic system of Xue et al., (2021) with a single convergence-divergence mixing and incubation channel. (**b**) Sandwich type colorimetric nanozyme-linked immunosorbent assay mechanism of the microfluidic system of Xue et al., (2021).

**Figure 12 ijms-23-04638-f012:**
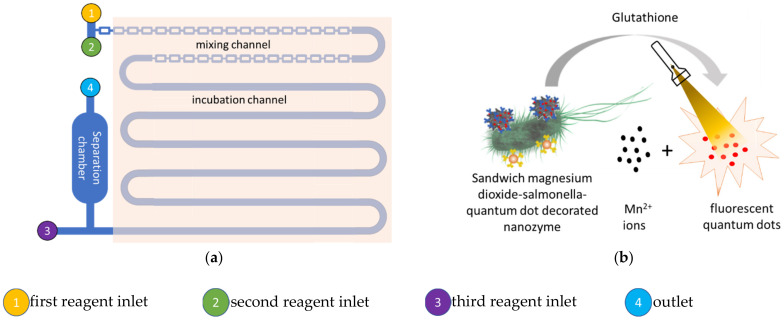
Illustrations of the colorimetric and fluorometric microfluidic systems by Xue et al., (2021) and Hao et al., (2020) using the manganese dioxide nanoflower nanozyme. (**a**) The nanozyme enhanced microfluidic system reported by Hao et al., (2020) consists of an initial mixing followed by an incubation channel. (**b**) Sandwich type fluorometric nanozyme-linked immunoassay of the microfluidic system reported by Hao et al., (2020).

**Figure 13 ijms-23-04638-f013:**
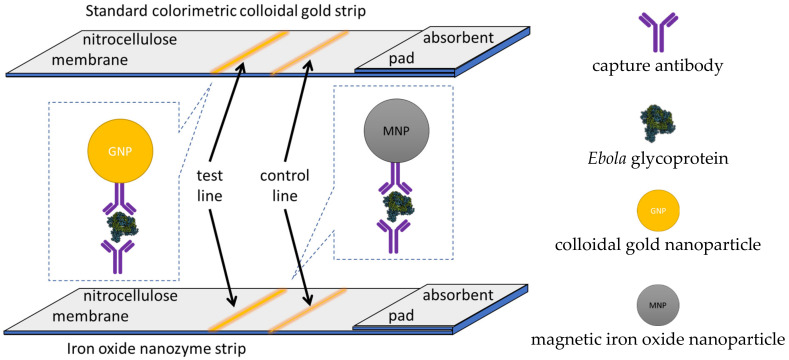
The iron oxide colorimetric immunoassay nanozyme strip developed for the indirect detection of *Ebola* and the standard strip made of colloidal gold nanoparticles.

**Figure 14 ijms-23-04638-f014:**
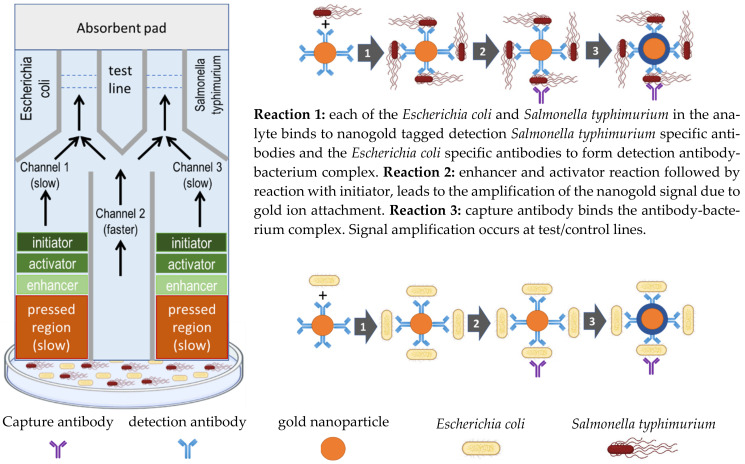
The operational principle of the paper-based nanogold-linked immunoassay dipstick reported by Park et al., (2016) showing the multi-speed multi-channel design. The pre-immobilized capture antibodies at the test line give the gold ion amplified signal when the gold ions are attached to the mobilized nanogold of the nanogold tagged detection antibody with pathogens from the fast flow rate channel 2.

**Figure 15 ijms-23-04638-f015:**
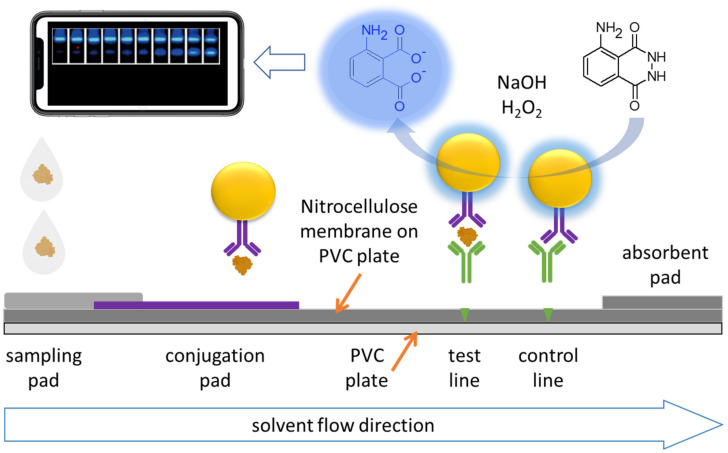
Illustration of the chemiluminescence nanozyme-linked immunoassay paper strip developed and tested by Liu et al., (2020) in the detection and identification of the SARS-CoV2 S-RBD antigen.

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
