# Peer review of "Applications of Nanozymology in the Detection and Identification of Viral, Bacterial and Fungal Pathogens"

_ijms, 2022, doi:10.3390/ijms23094638_

Round 1
Reviewer 1 Report
The review describes the nanoenzymological methods and approaches to detect and identify different pathogens. The review is organized greatly with both texts and illustrations. The overall methodology reflects broad applications in the medical and environment. The Authors have thoroughly analyzed the existing literature and specifically mentioned the related points with proper explanations. However, a few points are to be taken into consideration to enrich with more factual information in the line of the topic. Below mentioned comments are made for improving the current version of the manuscript:
Major comments:
- Page-2, Line no-69-72. “Of the six classes of agents ……… by viruses [14,15], bacteria [16,17], fungi [18,19], and parasites [20,21].” after T4. This sentence needs to be reconstructed to make more meaningful. Please write “pathogens” or “pathogenic agents” in place of “agent”.
- Page-5, Line no-172-175. “Bacillus subtilis, Streptococcus mutans, Pseudomonas aeruginosa, Staphylococcus aureus, Escherichia coli, and Salmonella enterica”. Species should be in italics throughout the manuscript. Please modify accordingly.
- Page-8, Line no-273-274. “In some of these studies the nanozymes are augmented with biological material such as enzymes, DNA and RNA strands or fragments.”. Author is requested to cite suitable references.
- Page-9, Line no-304. “4. Current status of pathogen detection”. Please write “Recent approaches” in place of “Current status”.
- Page-11, Line no-399. Please mention in bracket “GQD” after “graphene quantum dots”. All abbreviations should be explained properly throughout the manuscript.
- Figure 1. Legend- “there are two main classes of viruses, the isometric class and the bacteriophages”. It does not reflect the main classes. However, the author can write “Two major morphological classes………”.
- Figure 3. The graphical illustration is specifically represented for the detection of viruses. Is it possible to make it universally for all antigens? Please modify accordingly.
- Figure 4. Please specify “gold” in the figure for better understanding.
- The Authors are encouraged to write future implications with therapeutics and applications in genetic engineering and biotechnology.
Minor comments:
- Page-8, Line no-288. Please correct “enzymelike”.
Author Response
Major comments:
Page-2, Line no-69-72. “Of the six classes of agents ……… by viruses [14,15], bacteria [16,17], fungi [18,19], and parasites [20,21].” after T4. This sentence needs to be reconstructed to make more meaningful. Please write “pathogens” or “pathogenic agents” in place of “agent”.
Thanks. This has been done. With the references omitted, the sentence now reads as follows:
Among the six classes of pathogenic agents responsible for communicable diseases which include prions, viruses, bacteria, fungi, protozoa, and helminths, much research is establishing new technologies to diagnose infections caused by viruses, bacteria, fungi, and parasites.
Page-5, Line no-172-175. “Bacillus subtilis, Streptococcus mutans, Pseudomonas aeruginosa, Staphylococcus aureus, Escherichia coli, and Salmonella enterica”. Species should be in italics throughout the manuscript. Please modify accordingly.
Thanks. This has been done. All microorganism names have been italicised
Bacillus subtilis, Streptococcus mutans, Pseudomonas aeruginosa, Staphylococcus aureus, Escherichia coli, and Salmonella enterica.
Page-8, Line no-273-274. “In some of these studies the nanozymes are augmented with biological material such as enzymes, DNA and RNA strands or fragments.”. Author is requested to cite suitable references.
Thanks. The following two references have been cited to illustrate the point being made made.
Mao MX, Zheng R, Peng CF, Wei XL. DNA-Gold Nanozyme-Modified Paper Device for Enhanced Colorimetric Detection of Mercury Ions. Biosensors (Basel). 2020;10(12):211. Published 2020 Dec 18. doi:10.3390/bios10120211
Chen, W., Fang, X., Ye, X. et al. Colorimetric DNA assay by exploiting the DNA-controlled peroxidase mimicking activity of mesoporous silica loaded with platinum nanoparticles. Microchim Acta 185, 544 (2018). https://doi.org/10.1007/s00604-018-3026-9
Page-9, Line no-304. “4. Current status of pathogen detection”. Please write “Recent approaches” in place of “Current status”.
Thanks. This has been done. The line now reads: Recent approaches of pathogen detection
Page-11, Line no-399. Please mention in bracket “GQD” after “graphene quantum dots”. All abbreviations should be explained properly throughout the manuscript.
Thanks. This has been done throughout the manuscript.
Figure 1. Legend- “there are two main classes of viruses, the isometric class and the bacteriophages”. It does not reflect the main classes. However, the author can write “Two major morphological classes………”.
Thanks. The line now reads thus:
Figure 1: there are two major morphological classes of viruses, the isometric class and the bacteriophages
Figure 3. The graphical illustration is specifically represented for the detection of viruses. Is it possible to make it universally for all antigens? Please modify accordingly.
Yes. Thanks. This is indeed a universally applicable methodology.
The changes have been made in the diagram itself and the legend now also reads thus:
Figure 3: graphical illustration of generic microbial nanozyme-linked immunoassay methodology
Figure 4. Please specify “gold” in the figure for better understanding.
Yes. Thanks. This has been done. The gold working electrode is now clearly labelled “gold working electrode”.
The Authors are encouraged to write future implications with therapeutics and applications in genetic engineering and biotechnology.
This is an excellent idea. Thanks. A section 15 describes future implications with therapeutics and applications in genetic engineering and biotechnology has been included.
Please see section 15, lines 786-811

Reviewer 2 Report
“Applications of Nanozymology in the Detection and Identification of Viral, bacterial and Fungal Pathogens” by Sandile Phinda Songca
Nanozymes are synthetic nanoparticulate materials which mimic the biological activities of enzymes by virtue of their surface chemistry.
This review focused on the applications of nanozymes in the detection and identification of pathogens (viral, bacterial, fungal pathogens) and some application in cancer cell and parasite detection). in samples obtained from foods, natural environments, and clinical sources.
The author describes how nanozymology offers significant improvements in the six methodological indicators that are proposed as being key in this review, including simplicity, sensitivity, speed of response, cost, reliability, and durability of the immunoassays and polymerase chain reaction strategies.
Various systems are presented such as Direct Nanozyme-based Cancer Cell Detection, Bacterial detection and identification, Pathogen biosensors, Microfluidic biosensors, The paper dipstick or spot test methods, Different Flow Rate Multiple Channel Colorimetric Nanogold Paper Strip and
Chemiluminescence Nanozyme-Linked Immunoassay Paper Strip
Major and minor modifications need to be addressed in order to make this revew more aceesible to the readers.
1) Most probably, the readers of this review will not be specialized in the field of nanozymes therefore more detailed should be given in order to understand the basis. A minimum of explaination is necessary including why do we need surrogate enzyme such as these nanozyme.
2) chapter 5-Nanozymology
The approch described lines 377-389 is not very well explained and rather confusing for an non-expert. A definition of aptamer which is the heart of the interaction between nanozymes and novovirus should be given. What is the aptamer surrogate?? How this aptamer is produced and what is the structure of this aptamer? A figure is necessary
3) legend of Figure 4 should be more detailed with a precise descripton of all the steps. What is the exact mechanism?
4) A general comment : The amenability of the nanozymes to work in tandem with aptamers is not explained at all in this review and should be presented What is an aptamer ? and what is the purpose of the use of these surrogate ? nanozymes and aptamer based biosensing should be clearly explained .
5) No reference in the text to figure 5
6) chapter 6-Indirect Nanozyme-based Fungal detection
Nanozymes are implicated in then strategies for the destruction of fungal pathogens
Once more the mechanism of action of the nanozymes is not clear Lines 416 to 423 are very confusing and need a proper figure
8) Line 480 « detection of ova in the water » ova from which parasite ?
9) Line 493 A pellet obtained from the incubation of casein capped nanogold Why casin is used here ?
10) chapter Perspectives is just a repeat of the introduction. New strategies and dircetion should be described.
11) “PCR” should be used instead of “polymerase chain reaction” each time it is referred to this technique .
12) What is the enzymatic activity of these nanozymes? peroxydase activity is mentionned bur as a general question what are the other enzymatic activities of Nanozymes.
Author Response
|
Reviewer |
1) Most probably, the readers of this review will not be specialized in the field of nanozymes therefore more detailed should be given in order to understand the basis. A minimum of explanation is necessary including why do we need surrogate enzyme such as these nanozyme. |
|
Response |
This comment is highly appreciated as it is correct. In this review nanozymes have been defined and explained in various parts of the review, |
|
Reference |
In the abstract: “Nanozymes are synthetic nanoparticulate materials which mimic the biological activities of enzymes by virtue of their surface chemistry” |
|
Reference |
In the abstract: “Unlike the enzymes however, nanozymes are cost-effectively prepared, purified, stored, reproducibly and repeatedly used for long periods of time” |
|
Reference |
In the introduction: “One of the fastest growing applications of nanotechnology today is the identification of pathogens and diagnosis of the diseases they cause, is nanozymology, which is an emerging terminology used to refer to the science of nanozymes. Nanozymes are defined as nanomaterials which act like enzymes without most of the limitations of real enzymes; they are cost effectively and exogenously generated to catalyze many biological processes, yet they are stable enough for a long shelf-life storage. For example, iron oxide nanoparticles can act as surrogates to mimic the traditional peroxidase enzymes. Research on nanozymes has increased exponentially over the past decade and with this, the new interdisciplinary area of nanozymology was born” |
|
Response elaboration |
Specifically in the above definition found in the introduction, the reasons why we need surrogate enzyme such as these nanozymes is because they provide alternatives to natural enzymes due to the fact that they are cost effectively and exogenously generated to catalyze many biological processes, and that they are stable enough for a long shelf-life storage. |
|
Response elaboration |
To maintain brevity and avoid reader fatigue I beg leave not to expand beyond the foregoing. |
|
Reviewer |
2) chapter 5-Nanozymology The approach described lines 377-389 is not very well explained and rather confusing for an non-expert. A definition of aptamer which is the heart of the interaction between nanozymes and novovirus should be given. |
|
Response |
Thanks. A definition has been given. |
|
Reviewer |
2) chapter 5-Nanozymology The approach described lines 377-389 is not very well explained and rather confusing for an non-expert. What is the aptamer surrogate?? |
|
Reviewer |
Sorry about that. This has been removed. |
|
Reviewer |
2) chapter 5-Nanozymology The approach described lines 377-389 is not very well explained and rather confusing for an non-expert. How this aptamer is produced and what is the structure of this aptamer? |
|
Response |
Thanks. An elaborate answer is given in lines 380-411 |
|
Reviewer |
2) chapter 5-Nanozymology The approach described lines 377-389 is not very well explained and rather confusing for an non-expert. A figure is necessary. |
|
Response |
Thanks. Figure 3 has been provided. |
|
Reviewer |
3) legend of Figure 4 should be more detailed with a precise description of all the steps. |
|
Response |
This has been done. It is now figure . |
|
Reviewer |
3) What is the exact mechanism? |
|
Response |
The mechanism is given in line 421-434 |
|
Reviewer |
4) A general comment: The amenability of the nanozymes to work in tandem with aptamers is not explained at all in this review and should be presented What is an aptamer? and what is the purpose of the use of these surrogates? nanozymes and aptamer based biosensing should be clearly explained. |
|
Response |
Thank you. In response of this general comment a brief explanation was given. Please see chapter 5. Nanozymology. Lines 381-421. |
|
Reviewer |
5) No reference in the text to figure 5 |
|
Response |
Thanks. It is now Figure 6. Chemical structures of the fungal extract zearalenone, an estrogenic metabolite produced by the fungal species Gibberella zeae (Sun et al, 2018) [79]. |
|
Reviewer |
6) chapter 6-Indirect Nanozyme-based Fungal detection Nanozymes are implicated in the strategies for the destruction of fungal pathogens. Once more the mechanism of action of the nanozymes is not clear Lines 416 to 423 are very confusing and need a proper figure. |
|
Response |
Thanks. The mechanism is now explained in lines 454-461. |
|
Reviewer |
8) Line 480 « detection of ova in the water » ova from which parasite? |
|
Response |
Thanks. This is now clarified in lines 518-520 |
|
Reviewer |
9) Line 493 A pellet obtained from the incubation of casein capped nanogold Why casein is used here? |
|
Response |
Thanks. This is now clarified in lines 533-538 |
|
Reviewer |
10) chapter Perspectives is just a repeat of the introduction. New strategies and direction should be described. |
|
Response |
This is done in section 15. Future implications with therapeutics and applications In line with the recommendation of reviewer 1 a section on future perspectives discussing possible future implications with therapeutics and applications although this is slightly outside the intended scope of the paper. |
|
Reviewer |
11) “PCR” should be used instead of “polymerase chain reaction” each time it is referred to this technique . |
|
Response |
Except the first mention. all polymerase chain reaction is replaced by “PCR” |
|
Reviewer |
12) What is the enzymatic activity of these nanozymes? peroxydase activity is mentionned but as a general question what the other enzymatic activities of Nanozymes are. |
|
Response |
Focusing on the nanozymes which are discussed in this paper, three enzymatic activities of nanozymes are discussed in lines 167-171 |

Round 2
Reviewer 2 Report
All the points that I made have been addressed